# Adherence and Effects Derived from FODMAP Diet on Irritable Bowel Syndrome: A Real Life Evaluation of a Large Follow-Up Observation

**DOI:** 10.3390/nu12040928

**Published:** 2020-03-27

**Authors:** Antonietta Gerarda Gravina, Marcello Dallio, Mario Romeo, Antonietta Di Somma, Gaetano Cotticelli, Carmelina Loguercio, Alessandro Federico

**Affiliations:** Hepatogastroenterology Division, Department of Precision Medicine, University of Campania Luigi Vanvitelli, via Pansini 5, 80131 Naples, Italy; marcello.dallio@gmail.com (M.D.); marioromeo@virgilio.it (M.R.); antonietta-disomma@libero.it (A.D.S.); gaetano.cotticelli@unicampania.it (G.C.); carmelina.loguercio@unicampania.it (C.L.); alessandro.federico@unicampania.it (A.F.)

**Keywords:** IBS, low FODMAPs, diet, low gluten diet

## Abstract

Introduction: Irritable bowel syndrome represents one of the most difficult gastroenterological diseases to treat, that usually induces the patients to follow different drug therapies, often not useful in symptom control. In this scenario low FODMAP diet could have positive effects in patients with irritable bowel syndrome, even because this type of diet regimen is characterized by a low gluten amount due to the exclusion of cereals. Methods: We enrolled 120 patients with irritable bowel syndrome, according to the Rome IV criteria, who were referred to Hepatogastroenterology Division of the University of Campania L. Vanvitelli from June to December 2018. They underwent a low FODMAP diet for six weeks, followed by a gradual weekly reintroduction of every category of food for three months. The patients had a follow-up evaluation for six months after the end of food reintroduction period. We measured abdominal pain with subjective numerical scale from 0 to 10. We evaluated other gastrointestinal symptoms with a questionnaire about symptoms of lower digestive tract, evaluating their frequency and intensity. We also evaluated the impact of irritable bowel syndrome on daily life with neurological bowel dysfunction score. Results: We obtained a good patient-adherence to diet and a statistically significant decrease of abdominal pain, bloating, flatulence, diarrhea, constipation, and neurological bowel dysfunction score (*p* < 0.001) at the end of the diet. These results remained constant in the follow-up period. Conclusions: We recommend the use of a low FODMAP diet regimen in patients with irritable bowel syndrome in order to control the symptoms and improve the quality of life.

## 1. Introduction

The worldwide prevalence of irritable bowel syndrome (IBS) is on average of 10% and, specifically, 12% in Italy [1]. The prevalence in adults and adolescents has increased by 15–20%, whereas it is lower in elderly people, in which it is approximately 10%, with a female (F):male (M) ratio of 2:1 [1]. IBS implies high socio-economic costs in terms of medical examinations, unnecessary diagnostic investigations, medical treatments, and sometimes, even surgical interventions which are completely inappropriate. Therefore, this syndrome represents one of the most important socio-economic health problems [2]. IBS etiopathogenesis, as the greater part of functional gastrointestinal disorders, is not completely known. However, it has been hypothesized that its symptoms are caused by several concomitant factors. More precisely, three alterations, visceral hypersensitivity, defined as an increase in the perception of stimuli from the digestive tract, alterations in gastrointestinal motor activity and psychological stress, represent the main pathogenetic mechanisms underlying this syndrome [3,4,5]. Recent studies have demonstrated some alterations of the gut microbiota, immune system, and intestinal permeability in these patients; however, further studies are necessary in order to confirm a specific role of these factors in the disease development [6,7,8,9,10]. IBS diagnosis is based on the anamnesis, physical examination, and symptoms reported in accordance with the Rome IV criteria. The Rome IV criteria are recurrent abdominal pain, on average, at least 1 day per week in the last 3 months, associated with two or more of the following criteria: (1) relationship with defecation, (2) change in frequency of evacuation, (3) change in form (appearance) of stool. [1]. Usually the symptoms associated with IBS are abdominal pain and bloating associated with diarrhea (IBS-D), constipation (IBS-C), or both of them (IBS-M) [11]. There are different conditions that can be confused with IBS, including chronic inflammatory bowel diseases, systemic hormonal disorders, enteric infections, colorectal cancer, and malabsorption diseases [6,11]. Furthermore, symptoms such as rectal bleeding, iron deficiency anemia, weight loss, and a family history of gastrointestinal diseases should be considered, because their presence in a typical IBS patient is not usually so frequent [11]. Various medications are available to treat IBS symptoms, including antispasmodics, antidiarrheal, cathartic, motility agents, and antidepressants [12]. However these drugs are not often able to relieve the symptoms and for this reason some people prefer to avoid them by choosing alternative approaches [13]. Food has always had a significant role in IBS. However, the studies that explored the role of food in this context were not precise and included a reduction of milk and dairy products, avoiding large quantities of fruit and raw vegetables and/or a large consumption of fibers. Other suggested interventions include psychotherapy, cognitive-behavioral treatment, hypnosis, relaxation therapy, nutraceuticals, and acupuncture [7,12,13,14,15,16].

FODMAP is the acronym of “Fermentable Oligo-, Di- and Mono-saccharides and Polyols”. The term FODMAP includes sugars with a high fermentative power, particularly concentrated in several categories of food. They can be related to intestinal symptom appearance: swollen belly, meteorism, abdominal pain, diarrhea or constipation when consumed in significant quantities and regularly by sensitive subjects. These low-absorbed short-chain carbohydrates are able to exert an osmotic action, because they are poorly absorbed and rapidly fermented by bowel bacteria. This process causes an accumulation of water and gas in the intestinal lumen with a consequent distension of the bowel, that in turn determines, in a patients with visceral hypersensitivity, typical IBS symptom appearance [17,18,19]. Low-FODMAP diet, (i.e., a low oligosaccharide, disaccharide, monosaccharide and polyol diet) is associated with an improvement of gastrointestinal symptoms in 50–80% of patients [20,21,22]. This diet expected the elimination of wheat, barley, spelt, rye, and all the other gluten containing cereals, for a limited period of time, [22,23]. It is also important to highlight that these cereals also contain fructans, which in turn could be the responsible for the triggering of irritable bowel syndrome related symptoms. A clinical trial regarding the effect of low FODMAP diet in patients with IBS demonstrated the possibility to use this diet regimen as an effective therapy in symptoms control [24].

We therefore evaluated the adherence and long-term effects on IBS symptoms derived from the use of a low FODMAP diet for six weeks, followed by a gradual reintroduction of food categories.

## 2. Materials and Methods

A total of 120 consecutive patients with diagnosis of IBS according to the Rome IV criteria, who were referred to the Hepatogastroenterology Unit of University of Campania “L. Vanvitelli” for gastroenterological consultation between June and December 2018, were enrolled in the present study. Inclusion criteria were: age between 18 and 65 years; a negative colonoscopy for pathological findings. Exclusion criteria included a coexistence of the following conditions: diabetes, obesity, metabolic syndrome, chronic renal failure (CRF), thyroid disease, significant gastrointestinal diseases (e.g., inflammatory bowel diseases (IBD), celiac diseases, infectious colitis, gastrointestinal resection), lactose intolerance, psychiatric diseases, rheumatological diseases, use of drugs that can alter intestinal motility or probiotics or antibiotics. Patients followed a low FODMAP diet for six weeks with subsequent weekly reintroduction of individual categories of food for 3 months. They were followed-up for another 6 months (Figure 1).

A gastroenterologist gave dietary instructions to patients. This prospective study is in adherence with ethical guidelines of the Declaration of Helsinki (1975). Written informed consent was obtained from each enrolled patient before starting the diet.

All patients had a clinical assessment and symptomatic evaluation questionnaire at T0 (baseline), T1 (six weeks), T2 (three months from T1), T3 (six months from T2). During the study period, patients were advised not to take medications that could interfere with their assessment.

### 2.1. Low-FODMAP Diet

The fermentable carbohydrates identified until now and food that contains them are:Fructans, fructose-containing oligodisaccharides, abound in cereals, some vegetables and fruits and legumes;Galacto-oligosaccharides, polymers consisting of galactose, fructose and glucose;Lactose, the disaccharide typical of milk and its fresh derivatives;Fructose, a monosaccharide particularly concentrated in some fruits, in honey and vegetables. It could be responsible for health and gastrointestinal symptoms appearance, particularly when it is introduced with high amount of glucose that in turn is responsible for its absorption;polyols, monosaccharides such as sorbitol and xylitol, used as sweeteners and humectants in food industry and abundant in hard-core fruits and some vegetables (Table 1).

Quinoa and carrots are types of food with a reduced FODMAP content that can be used without restrictions in the elimination diet.

### 2.2. Reintroduction Diet

There are no clear indications in scientific literature regarding the order with which to test the effects of various FODMAPs reintroduction. It could be made on the basis of personal experiences, tastes and preferences. In the present study we used the following succession:Fructans: it should start with one or two slices of white bread or about thirty grams of pasta. The onion represents a vegetable rich in fructans: in this case it is advisable to consume at maximum half an onion, that corresponds to about ten grams, per test;Lactose: to start with half a glass of milk or a packet of plain yogurt;Fructose: to start with half an apple or a teaspoon of honey;Polioli: regarding the sorbitol to start with two apricots or half a peach; for the mannitol test with 100 g of fresh mushrooms;Galacto-oligosaccharides: to start with 100 g of lentils or boiled chickpeas.

If the reintroduction of a specific FODMAP did not cause symptoms it was possible to test other food containing the same FODMAP, or to pass to evaluate a new FODMAP, considering the indications on the test modality. If the reintroduction was accompanied by the reappearance of symptoms, then patients could proceed as follows:-Return to a low FODMAP diet and once the symptoms have disappeared test again the food that gave problems but starting from a halved portion;-If the symptoms appear again, the tested FODMAP probably causes problems even with reduced portions and frequencies. To verify this hypothesis, it was possible:To test another food that contains the same FODMAP;To test again after a period of strict diet;To test again in the future to confirm the problems encountered.

### 2.3. Evaluation of Symptoms and Adherence to Diet

To evaluate symptoms we used the following questionnaires: (1) Neurological bowel dysfunction (NBD) score [25]: it provides a score from 0 to 47 (0–6 very low dysfunction, 7–9 low dysfunction, 10–13 moderate dysfunction, ≥14 severe dysfunction); (2) evaluation of abdominal pain using a 1 up to 10 levels scale of intensity (no pain: score 0–1, mild pain: score 2–3, average pain: score 4–6, severe pain: score 7–8, unbearable pain: score 9–10); (3) questionnaire about gastrointestinal symptoms of the lower digestive tract, abdominal bloating, flatulence, abdominal pain, constipation, diarrhea and stool consistency by using the Bristol scale [26]. Every symptom was evaluated considering the frequency (0: never; 1: less than 1 episode per week; 2: less than 3 episodes per week; 3: greater than three episodes per week; 4: daily) and the intensity (0: absent, 1: mild (does not interfere with daily activity), 2: moderate (limiting daily activity), 3: severe (which impedes daily activity) [27]. Adherence to the diet was evaluated at T1 and T2 by original Morisky Scale consisting of four questions. Each positive answer corresponded to the number 1 and each negative answer corresponded to the number 0. Patients with a score of 0–2 were considered non-adherent to the diet; on the contrary, those with a score of 3–4 were considered adherent [28].

### 2.4. Statistical Analysis

All data were analyzed using STATA version 10 for MacIntosh. Abdominal pain was recorded by 10 levels visual analog scale (VAS). We also recorded the frequency of symptoms classifying the patients into five classes (0: never; 1: less than 1 episode per week; 2: less than 3 episodes per week; 3: greater than three episodes per week; 4: daily). Normality testing was performed, and the symptoms data were not normally distributed. To analyze the change in symptom severity, the individual scores for each symptom were added and the means calculated. The comparison of continuous/ordinal variable between two different time of observation was assessed using the non-parametric Wilcoxon signed ranks test. We considered a *p*-value lower than 0.05 as significant. 

## 3. Results

We enrolled a total of 120 patients with IBS diagnosis (72 F, 48 M, mean age 41.2 years, age range 20–65) from June to December 2018 and we followed them in a six-month follow-up period since the end of the reintroduction diet period. A total of 120, 112, 108, and 100 patients were analyzed at the specific time point of the study: T0 (baseline), T1 (end of six weeks of low-FODMAP diet), T2 (end of three months-reintroduction diet), and T3 (end of six months-follow up period) respectively (Table 2).

The enrolled patients showed an important clinical improvement achieved by dietary treatment. By analyzing the evaluation questionnaires, we observed a significant improvement obtained at the end of the Low-FODMAP diet. This improvement was maintained during the follow-up period. Specifically, we observed a marked improvement in abdominal pain, assessed by using the numerical scale from 0 to 10, that showed a statistically significant variation at the end of restriction diet (T1), at the end of reintroduction period (T2), and at the end of follow-up period (T3) if compared to the baseline (T0), *p* < 0.05 (Table 3).

An improvement of the impact that IBS has on the daily life of the enrolled subjects, evaluated by NBD score. There was a significant reduction (*p* < 0.05) in the total NBD-score, from a moderate/severe score (≥10) to a lower score (<10) in most patients, with a statistically significant reduction of the mean score at the end of restriction diet (T1), at the end of reintroduction period (T2) and at the end of follow-up period (T3) than baseline (T0) (Table 3). 

We performed an evaluation of adherence to the diet through original Morisky Scale [28] and only eight patients did not follow completely the restriction diet at T1 (Table 3). Patient who was not adherent to the diet at T1 was lost to follow-up at T2. In addition, other four patients were not adherent to reintroduction diet and we lost them at T2. All other patients, at T1 and T2, had a score of 3–4 were considered adherent to diet. At T3 only 100 patients completed final evaluation. Complete adherence was 112/120 (93.3%) at T1, 108/120 (90%) at T2. Three patients on the reintroduction diet showed a recurrence of abdominal pain and flatulence when they re-introduced rich in oxalate food if they ate them in quantities greater than the recommended ones (one patient had problems eating more than 100 g of red beets, one patient eating more than 30 g of hazelnuts, and another one had problems eating more than 50 g of white or milk chocolate). The first two patients eliminated this food and once the symptoms disappeared they tested again it. In particular, they tested a halved portion and they did not show a recurrence of symptoms anymore. The last patient eliminated white or milk chocolate because he had symptoms despite the halved portion testing. All three patients were adherent to reintroduction diet.

A statistically significant improvement of abdominal bloating, flatulence, and abdominal pain, assessed by the evaluation questionnaire, was obtained both in terms of frequency and severity (Table 4 and Table 5). Furthermore, in IBS patients with diarrhea, a reduction in the frequency and severity of diarrheal episodes, the main cause of social distress in these patients, was also highlighted. Patients who followed the Low-FODMAP diet showed a tendency to “normalize” the fecal consistency, with marked results especially in subjects reporting a Bristol consistency scale degree between 3 and 5 at baseline [26].

## 4. Discussion

Worldwide IBS is one of the most common pathologies of the gastrointestinal tract, as well as one of the main causes of high healthcare costs [2]. Most patients recurred to symptomatic drug treatments. In the first instance, they are often ineffective or impossible to use for a long-term period avoiding the drug related side effects. In the recent past the low FODMAPs diet has been proposed as a possible first line of treatment in IBS patients. In scientific literature a lot of studies demonstrated that the low-FODMAP diet could be effective in IBS-related gastrointestinal symptom improvement [29,30,31]. As already extensively described, the therapeutic approach of low-FODMAP diet consists of two phases: the “elimination” and the “reintroduction.” In the first phase, patients have to follow a restrictive dietary regimen aimed at eliminating all the food containing the so-called FODMAPs. In the second phase, each category of food, previously eliminated, is reinserted gradually in the daily dietary regimen of these patients, in order to identify those categories mainly responsible for the recurrence of gastrointestinal symptoms. The adherence to the diet is a difficult medical challenge because of the fact that the rapid change of dietary behavior, typically associated with western diet, discourages and often demotivates patients to follow carefully the instructions provided. In our low Fodmap diet regimen some moderate source of FODMAPs, such as buckwheat and grapefruit, are lacking. We preferred to allow the use of some moderate low FODMAPs food that is not usually used in daily routine, in order to increase the compliance to the diet. The Morisky scale [28] used in our study demonstrated that a careful coordination allowed most patients to be able to follow strictly our instructions. To achieve this goal it was crucial to explain to each patient the progress of the various phases of the diet, keeping them informed of each step. Above all, we observed how a good doctor–patient relationship is essential for this type of treatment. We gave the dietary instruction during a specialistic gastroenterological check-up. This could be considered a limitation of the study because dietitians usually gave instructions regarding the diet in routine clinical practice. Another limitation is the lack of reported intake of energy, carbohydrates, fat, and protein during the diet. Adherence to the diet was, however, ≥90%, confirming the data in the literature [30,31]. This good adherence rate can be explained because of the progressive improvement of IBS-related symptoms obtained owing to the low-FODMAP diet. The lack of adherence in the patients was due to practical reasons, such as the lack of time to organize the daily food plan and in some cases, the poor response of the gastrointestinal symptoms to the same diet. Our study confirms that the low-FODMAP diet could be useful to obtain a long-term relief of IBS symptoms, in accordance to the data present in scientific literature about this topic [20]. On the contrary, other studies found a short time improvement in gastrointestinal symptoms [32,33,34]. Our study showed a significant improvement in all the evaluated symptoms and these results remained constant during the follow-up period of six months in the greater part of the enrolled patients. These data are in agreement with a lot of clinical studies [18,34,35,36]. We demonstrated an important reduction of abdominal pain in our patients. This reduction was present at the end of the restrictive diet, until the end of the reintroduction period and the end of the follow-up period. We can explain this result because patients with IBS usually have a surrounding visceral hypersensitivity and are more susceptible to bowel distension. Low FODMAPs diet reduces the amount of bowel gas, as demonstrated by Patcharatrakul T et al. and this, in turn, could be the reason of our observation [37]. They demonstrated a difference in H2 and CH4 breath concentrations, before and after low FODMAPs diet. There was a statistically significant H2 reduction after 4 weeks of low FODMAPs diet and a not statistically significant reduction of CH4 concentration evaluated at the same time point of the study.

One limitation of our study is the lack of a control group to compare the effect of the diet. All the evaluated symptoms during the follow-up period maintained the improvement registered at the end of the previous time points. This fact could be reasonably explained bearing in mind that the enrolled patients, during the phase of reintroduction diet, were able to identify the triggered foods for the symptom recurrence. In this way they drastically reduced or eliminated completely the specific food from their daily dietary regimen reaching a long term control of IBS symptoms.

Our study demonstrated an improvement in symptoms in all IBS variants, unlike some works that reported in the scientific literature a symptoms improvement only in patients with IBS-M and IBS-D variant [18,38,39,40].

However as proved by several studies reported in scientific literature low FODMAPs diet regimens could be associated with a reduction in calcium serum levels, as well as in the prevalence of beneficial bacteria in gut microbial population [38,39,40,41].

For this reason, before stating the low FODMAP for a period longer than 6 weeks, further studies are needed to prove the safety of this dietary regimen. Food associated with symptoms recurrence during the reintroduction phase was that rich in oxalate if eaten in larger quantities than the recommended ones. This is in agreement with a study by Nawawi et al. [19], but there are no clear explanations for this association. We can assert that the low-FODMAP diet is associated with a reduction in gastrointestinal symptoms in IBS-patients and that, as suggested by the guidelines of the National Institute for Health and Care Excellence [42], it is recommended to IBS-patients as advanced dietary therapeutic strategy, consequent to a modification of the lifestyle and a regulation of dietary behavior without major restrictions.

Further studies are needed in order to evaluate the long-term effects of the low FODMAPs diet and its effects on the gut microbiota composition, evaluating the possibility to use probiotics to prevent the low FODMAPs diet-associated dysbiosis [43].

## 5. Conclusions

Our study demonstrated that the low FODMAPs diet is useful for patients with IBS reducing the symptoms related to this clinical condition. This benefit persists both at the end of the restrictive diet period and at the end of the reintroduction period, but also at the end of a long term follow-up of 6 months. We need further studies with a long follow-up like our study or probably even longer and with a great number of enrolled patients to confirm our results and to assess the possibility to use this dietary regimen for more than 6 weeks without side effects.

## Figures and Tables

**Figure 1 nutrients-12-00928-f001:**
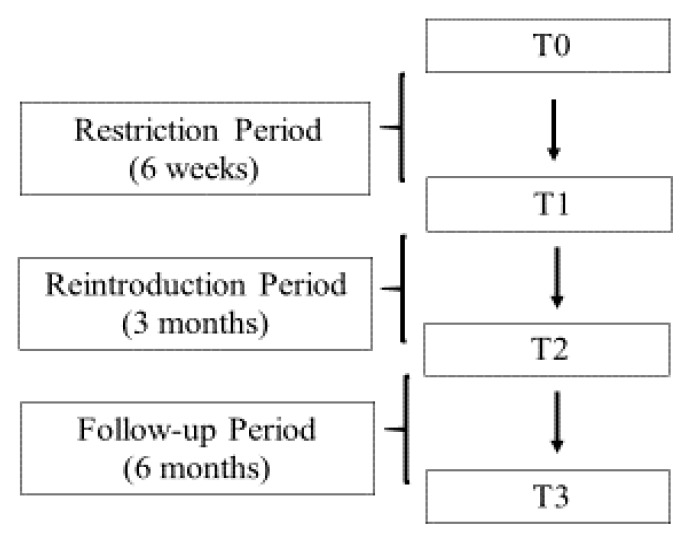
Flow diagram study design.

**Table 1 nutrients-12-00928-t001:** Description of high and moderate Fodmap containing foods.

Food with High Content of FODMAP (This Food Should not Be Consumed during Low-FODMAP Diet)	Food with Moderate Content of FODMAP (This Food Can Be Consumed in Small Quantities, Paying Attention not to Consume Other Food with Moderate FODMAP Content at the Same Time)
Dairy products: cream, ice cream, milk desserts, milk powder. Milk: fresh cow, goat, or sheep milk. Yogurt: cow’s milk yogurt, goat’s milk, or sheep’s milk Cheeses: ricotta, mascarpone, milk flakes, and all fresh cheeses Fruit: peaches, persimmons, watermelon, apples, pears, figs, cherries, apricots, plums, blackberries Vegetables: artichokes, garlic, onion, leek, shallot, asparagus, peas, broad beans, cabbage, broccoli, brussels sprouts, cabbage, mushrooms, peas Cereals: bread, pasta and wheat derivatives, especially if consumed in large quantities. Barley and derivatives, rye and derivatives Legumes: lentils, chickpeas, beans Oily dried fruit: pistachios, cashews Fibers and supplements: inulin, FOS (fructooligosaccharides) used in many foods marketed as prebiotics Vegetables: fennel, corn (50 to 100 g), celery (2–3 ribs), sweet potatoes (120 g). Sweeteners and additives: agave, glucose-fructose syrup (present in many carbonated beverages), fructose, fruit juices and fruit concentrates, sorbitol, mannitol, maltitol, isomalt, lactitol, xylitol, erythol	Milk: small amounts of milk added to coffee Dairy products: cream and butter Fruit: pomegranate (half fruit), grapefruit (half fruit), avocado (half fruit), cherries (max 5), litchi (max 5). Vegetables: fennel, corn (50 to 100 g), celery (2–3 ribs), sweet potatoes (120 g). Oily dried fruit: almonds, hazelnuts (about ten). Sweeteners: cooking sugar, molasses, maple syrup

**Table 2 nutrients-12-00928-t002:** Characteristics of the enrolled patients.

Characteristics	T0 (Baseline) *n* = 120	T1 (End of Six Weeks of Low-FODMAP Diet) *n* = 112	T2 (End of Three Months-Reintroduction Diet) *n* = 108	T3 (End of Six Months-Follow up Period) *n* = 100
Female	72	68	64	60
Male	48	44	44	40
IBS-C	24	24	24	20
IBS-D	28	28	24	24
IBS-M	68	60	60	56

**Table 3 nutrients-12-00928-t003:** Description of abdominal pain, neurological bowel dysfunction (NBD) score, and adherence at T0, T1, T2, and T3.

Used Scales	Mean Score at T0 (Baseline) ± SD	Mean Score at T1 (End of Six Weeks of Low-FODMAP Diet) ± SD	Mean Score at T2 (End of Three Months-Reintroduction Diet) ± SD	Mean Score at T3 (End of Six Months-Follow up Period) ± SD	*p*
Abdominal pain	9.1 ± 0,71	1.643 ± 1.06	2.111 ± 1.52	2.680 ± 1.72	T1, T2 and T3 vs. T0 < 0.05 T3 and T2 vs. T1 n.s.
NBD score	12.8 ± 1.27	6.536 ± 0.83	6.741 ± 0.71	6.4 ± 0.86	T1, T2 and T3 vs. T0 < 0.05 T3 and T2 vs. T1 n.s.
Adherence		3.414 ± 0.68	3.464 ± 0.69	3.621 ± 0.65	T3 and T2 vs. T1 n.s.

**Table 4 nutrients-12-00928-t004:** Frequency of analyzed symptoms at T0, T1, T2, and T3.

Symptoms	Mean at T0 (Baseline) ± SD	Mean at T1 (End of Six Weeks of Low-FODMAP Diet) ± SD	Mean at T2 (End of Three Months-Reintroduction Diet) ± SD	Mean at T3 (End of Six Months-Follow up Period) ± SD	*p* *
Bloating	3.5 ± 0.50 (frequency)	1.464 ± 0.50 (frequency)	0.6296 ± 0.49 (frequency)	0.52 ± 0.50 (frequency)	T1, T2 and T3 vs. T0 < 0.05
Flatulence	3.667 ± 0.47 (frequency)	1.857 ± 0.35 (frequency)	0.7407 ± 0.44 (frequency)	0.68 ± 0.47 (frequency)	T1, T2 and T3 vs. T0 < 0.05
Diarrhea	2.223 ± 1.28 (frequency)	0.7857 ± 0.41 (frequency)	0.5185 ± 0.50 (frequency)	0.3600 ± 0.48 (frequency)	T1, T2 and T3 vs. T0 < 0.05
Constipation	2.1 ± 1.39 (frequency)	0.75 ± 0.44 (frequency)	0.5556 ± 0.50 (frequency)	0.5600 ± 0.56 (frequency)	T1, T2 and T3 vs. T0 < 0.05

Every symptom was evaluated for the frequency (0: never; 1: less than 1 episode per week; 2: less than 3 episodes per week; 3: greater than three episodes per week; 4: daily). A higher score means more severe symptoms. * *p* < 0.05.

**Table 5 nutrients-12-00928-t005:** Severity of analyzed symptoms at T0, T1, T2, and T3.

Symptoms	Mean at T0 (Baseline) ± SD	Mean at T1 (End of Six Weeks of Low-FODMAP Diet) ± SD	Mean at T2 (End of Three Months-Reintroduction Diet) ± SD	Mean at T3 (End of Six Months-Follow up Period) ± SD	*p* *
Bloating	2.6 ± 0.47 (severity)	1 ± 0.0 (severity)	0.62 ± 0.4921(severity)	0.52 ± 0.50 (severity)	T1, T2 and T3 vs. T0 < 0.05
Flatulence	2.4 ± 0.49 (severity)	1.143 ± 0.35 (severity)	0.7407 ± 0.44 (severity)	0.68 ± 0.47 (severity)	T1, T2 and T3 vs. T0 < 00.05
Diarrhea	1.200 ± 0.76 (severity)	0.7857 ± 0.41 (severity)	0.5185 ± 0.50 (severity)	0.3600 ± 0.48 (severity)	T1, T2 and T3 vs. T0 < 0.05
Constipation	1.733 ± 1.04 (severity)	0.75 ± 0.44 (severity)	0.5556 ± 0.50 (severity)	0.5600 ± 0.56 (severity)	T1, T2 and T3 vs. T0 < 0.05

Every symptom was evaluated for the intensity (0: absent; 1: mild (does not interfere with daily activity); 2: moderate (limiting daily activity); 3: severe (which prevents daily activity). A higher score means more severe symptoms. * *p* < 0.05.

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
