# Peer review of "Adherence and Effects Derived from FODMAP Diet on Irritable Bowel Syndrome: A Real Life Evaluation of a Large Follow-Up Observation"

_nutrients, 2020, doi:10.3390/nu12040928_

Round 1
Reviewer 1 Report
Compliance and effects derived from FODMAP diet on irritable bowel syndrome: a real life evaluation of large follow-up observation.
ABSTRACT: This manuscript was obviously translated into English and is difficult to read at times. Please check for English grammar and spelling (ie Conclusions: “raccommend” incorrectly spelled “recommend”. This was a huge turn off for the reviewer. The term “compliance” is generally not used anymore since it implies a passive behavior in which a patient is following instructions rather than a more positive proactive behavior, “adherence” which results in lifestyle changes by the patient implying the patient is taking a more active role in their care/treatment. In the abstract I am used to seeing statistical quantifications e.g. p = .02 to represent the statistical reduction in symptom reporting that is significant. Also, mentioning instrument used to measure symptoms or even what symptoms were measured would be beneficial. The abstract needs to be re-written to clearly provide information to entice the reader and to appeal to the readers interest that this is an actual scientific article.
INTRODUCTION:
L40 Define terms before using abbreviation e.g. f = female, m=male. Although it is assumed it is best practice to define ALL terms before using abbreviation the first time in the text.
L41 citation needed regarding “determine costs…”
L51 Diagnosis should not be capitalized
L52 Define Rome IV criteria
L53 there are other symptoms associated with IBS, swelling would not be the best choice of words here, consider bloating or distention. The way swelling is used implies inflammation not bloating a key feature of IBS
L62 Since your whole study is on FODMAPs I would provide far more detail regarding other studies and diet changes details. One line reference does not provide enough support for doing this study.
L76 this diet involves many other things than just excluding wheat, barley, spelt, rye. Additional reference to this would strengthen this section.
L79-80 you comment on no other prospective studies in literature that evaluate patients using low-FODMAP. Be careful when saying NO other literature out there. Here is one,
Neurogastroenterol Motil. 2018 Jan;30(1). doi: 10.1111/nmo.13154. Epub 2017 Jul 14.
Long-term impact of the low-FODMAP diet on gastrointestinal symptoms, dietary intake, patient acceptability, and healthcare utilization in irritable bowel syndrome.
O'Keeffe M1, Jansen C1, Martin L1, Williams M2, Seamark L2, Staudacher HM1, Irving PM1,3, Whelan K1,3, Lomer MC1,3.
METHODS: L93 Since this is a prospective study more detail regarding low-FODMAP instruction would strengthen the report. See below article that provides guidelines for instruction.
J Hum Nutr Diet. 2018 Apr;31(2):239-255. doi: 10.1111/jhn.12530. Epub 2018 Jan 15.
The low FODMAP diet in the management of irritable bowel syndrome: an evidence-based review of FODMAP restriction, reintroduction and personalisation in clinical practice.
Whelan K1, Martin LD1, Staudacher HM1,2,3, Lomer MCE1,2,4.
L146 Morisky Scale typically is used for medication adherence. Was this adapted for diet adherence? If so what is the reliability? Also you say each positive answer is score of zero and negative answer has a score of 1. Since there are only 4 questions listing an example or all 4 would help me understand better what is meant by positive and negative. Negative I think of nonadherence which would mean those with a higher score. I find this confusing. Can you provide examples and reliability of the questionnaire that you modified to gather this data?
L158 Why did you set p-value to 0.01? typically it is 0.05. Can you provide rationale for this please?
RESULTS:
L171-174 No need to repeat the numbers in the text. You can provide a verbal description of the findings and then refer to the data in the table 3. Pick one or the other to present the numerical data with the standard deviations and p-values. If keep table must add p-values
Tables 4 & 5 a note at the bottom reminding the reader of the range of scores e.g. higher score means more severe symptoms would be helpful. I couldn’t remember what a 0.6296 meant for bloating. It was reduced a lot but was it significant too? Add p-values to table. Or you can put an * next to the ones that are statistically significant. Plus it is recommended usually to go to two decimal places on standard deviation rather than four. On column headings add to your mean (±standard deviation) so it is clear to the reader this is what is listed. Tables need to be freely understood.
L193-198 So who was included in the analyses? All patients that where adherent?
Statistical analysis:
DISCUSSION:
L214 There are actually 3 stages. Restriction, reintroduction, and personalization (see above reference)
L249 can you provide any speculation why this is? And where future research needs to focus to clarify? The information that you have under conclusion should go into your discussion.
CONCLUSION:
The conclusion should be ONLY your conclusions on your study and how it contributes to the literature. You should not bring in new literature here. Move to your discussion and provide a greater detailed discussion.

Reviewer 2 Report
This prospective FODMAP study with re-introduction study contain some interesting data and is in some parts in line with previous studies. It is relatively large and emphasis has been put into re-introduction period. However, the text needs some improvements.
Minor comments:
- "Therefore this syndrome is one of the most expensive health problems". Is this truly correct, please give a proper citation.
- Who gave the dietary instructions, dietitian? IF not dietitian, ponder in limitations. You write in the discussion "To achieve this goal it was crucial to explain to each individual patient the progress of the various phases of the diet, keeping them informed of each step, but above all we observed how a good doctor - patient relationship is essential for this type of treatment."
- Re-Write lines 80-81, there seems to be at least one additional prospective FODMAP study with re-introduction period (Neurogastroenterol Motil. 2018 Jan;30(1).)
- Was the informed consent written one, and asked before commencement of the study?
- Add to the tables translation of time points T0 (Baseline), T1 (six weeks), T2 (three months, ... Tables should be self-explanatory.
- Line 106 fructose. Fructose is not a FODMAP as such but only in excess to glucose. Please, revise.
- Consider presenting symptom data as figures (lines), might be easier to read/follow.
- Line 152 "degrees" -> severity?
-
Just curious, why did you choose a p-value lower than 0.01 as significant and not conventional 0.05?
Major comments:
- English must be extensively revised.
- Instructions on low FODMAP diet. Did you provide information on which foods are allowed and recommended (low in FODMAP)? Patients often find it hard to conceive alternative food items in different food categories if not throughly addvised. Please include your written Italian instructions to patients as a supplement.
- Limitations of the study must be addressed fully and in detail, now lacking almost entirely.
- non-validated measurements of IBS symptoms. Why you did not choose IBS-SSS or GSRS which are validated gastro intstruments and widely used in IBS studies?
- First paragraph of the discussion is repetition, and belongs to the introduction/can be deleted. Please discuss your findings primarily and contrast to existing literature; get to the main points of your study quickly.
- Discuss placebo effect extensively which seems very high in your study. Especially abdominal pain was highest possible (Never seen before in any IBS study on average 9, on scale 0-10) and was reduced more than 70%. Frankly, this is unprecedented, almost sounds unbelievable and must be discussed, and reason explained. Also, low FODMAP diet does not reduce constipation in RCTs, in your study it did.
- Discuss your application of low FODMAP diet, it seems that some high and moderate sources of FODMAPs are lacking in your instructions. As an example, buckwheat and grapefruit are moderate in FODMAPs but lacking in your table. I'm not saying you did something wrong, rather that you did not institute low FODMAP diet in its strictest form; this deserves to be mentioned.
- Discuss what other things in diet may have changed in addition to amount of FODMAPs. You do not report intake of energy, carbohydrates, fat and protein. Did you measure them, if not, why not? If you did, please report.
- One can argue that the diet may have become a low calorie diet unintentionally, by doing so reduced symptoms. Did you weigh your patients at the start and at the end? If weight was reduced on average, this might explain extraordinary results.
- Did instruct restraining from probiotics, peppermint oil capsules, fiber supplements such as psyllium, or recommend reducing caffeine, bubbly drinks and fat consumption? Did you monitor the use of these possible dietary dietary treatment modalities?
- Your application of re-introduction is interesting. Please contrast it shortly to the methods recommended by Kings college and Monash team J Hum Nutr Diet. 2018 Apr;31(2):239-255.
Author Response
Response to Reviewer 2 Comments
Point 1 (Minor Comments):
"Therefore this syndrome is one of the most expensive health problems". Is this truly correct, please give a proper citation.
Who gave the dietary instructions, dietitian? IF not dietitian, ponder in limitations. You write in the discussion "To achieve this goal it was crucial to explain to each individual patient the progress of the various phases of the diet, keeping them informed of each step, but above all we observed how a good doctor - patient relationship is essential for this type of treatment."
Re-Write lines 80-81, there seems to be at least one additional prospective FODMAP study with re-introduction period (Neurogastroenterol Motil. 2018 Jan;30(1).)
Was the informed consent written one, and asked before commencement of the study?
Add to the tables translation of time points T0 (Baseline), T1 (six weeks), T2 (three months, ... Tables should be self-explanatory.
Line 106 fructose. Fructose is not a FODMAP as such but only in excess to glucose. Please, revise.
Consider presenting symptom data as figures (lines), might be easier to read/follow.
Line 152 "degrees" -> severity?
Just curious, why did you choose a p-value lower than 0.01 as significant and not conventional 0.05?
Response 1:
the following one is the added citation regarding “….most expensive health problems …”
Line 69à Chey, W.D.; Kurlander, J.; Eswaran, S. Irritable bowel syndrome: a clinical review. JAMA 2015, 313, 949–58.
L 329-331 A gastroenterologist gave dietary instructions and I wrote it in limitations
L 80-81àL 149-150 “There is one study about the effect of low FODMAP diet in patients with IBS in scientific literature”
L 173-174 Written informed consent was obtained from each enrolled patient before starting the diet.
Table 2, 3, 4,5 I added to the tables, the explanations of time points T0 (Baseline), T1 (six weeks), T2 (three months, ….... )
L106àL 185-186 I revised “fructose, a monosaccharide that abounds in some fruits, in honey and in some vegetables is a problematic sugar, but only when it is present in excess of glucose, with which it is absorbed;”
L152àL 236 "degrees" I changed it in Abdominal pain
I thank you for the suggestion in presenting the tables as figures, but another reviewer just, (you are reviewer 2), suggested me to leave the tables and remove the statistical data from the paper. I’ d like the editor decides how.
Line 243 We had a great number of patients and so we chose 0.01 to make the study more statistically significant, but I changed it with p< 0.05 because another reviewer asked me it.
Point 2 (Major comments):
Instructions on low FODMAP diet. Did you provide information on which foods are allowed and recommended (low in FODMAP)? Patients often find it hard to conceive alternative food items in different food categories if not throughly addvised. Please include your written Italian instructions to patients as a supplement.
I’ll include the written Italian instructions as attached files
Limitations of the study must be addressed fully and in detail, now lacking almost entirely.
Line 325 I wrote about this limitation
- non-validated measurements of IBS symptoms. Why you did not choose IBS-SSS or GSRS which are validated gastro intstruments and widely used in IBS studies?
We chose this measurements of IBS symptoms because these are measurements already used in other our studies and we are more familiar with these measurements
- First paragraph of the discussion is repetition, and belongs to the introduction/can be deleted. Please discuss your findings primarily and contrast to existing literature; get to the main points of your study quickly.
I modified Discussion section as your suggestions
- Discuss placebo effect extensively which seems very high in your study. Especially abdominal pain was highest possible (Never seen before in any IBS study on average 9, on scale 0-10) and was reduced more than 70%. Frankly, this is unprecedented, almost sounds unbelievable and must be discussed, and reason explained. Also, low FODMAP diet does not reduce constipation in RCTs, in your study it did.
Line 341-355 I discussed the reduction of abdominal pain: “We demonstrated a great reduction of abdominal pain in our patients. This reduction was present at the end of the restrictive diet and until both the end of the reintroduction period and at the end of the follow-up period. We can explain this great results because patients with IBS have visceral hypersensitivity and they are more susceptible to distension of the bowel due to the presence of bowel gas. Low FODMAPs diet reduces the amount of bowel gas, as demonstrated by Patcharatrakul T et al. They demonstrated difference in H2 and CH4 concentrations, evaluated by breath,before low FODMAPs diet and after it, because there was a statistically significant H2 reduction after 4 weeks of low FODMAPs diet and CH4 concentrations tended to be lower than those after restriction diet but this reduction did not reach statistical significance (Patcharatrakul T, Juntrapirat A, Lakananurak N, Gonlachanvit S.Effect of Structural Individual Low-FODMAP Dietary Advice vs. Brief Advice on a Commonly Recommended Diet on IBS Symptoms and Intestinal Gas Production. Nutrients. 2019 Nov 21;11(12). pii: E2856. doi: 10.3390/nu11122856).”
Line 362-364 we already discussed the reduction of constipation in our patients in contrast with other studies
- Discuss your application of low FODMAP diet, it seems that some high and moderate sources of FODMAPs are lacking in your instructions. As an example, buckwheat and grapefruit are moderate in FODMAPs but lacking in your table. I'm not saying you did something wrong, rather that you did not institute low FODMAP diet in its strictest form; this deserves to be mentioned.
L 322-325 I discussed this point “In our Low Fodmap diet some moderate source food of FODMAPs are lacking, such as buckwheat and grapefruit. We preferred to leave some moderate Low FODMAPs foods that are not usually used a daily routine in order to increase compliance with the diet which already in this way risked being limited because of the many limitations of the same diet.”
Discuss what other things in diet may have changed in addition to amount of FODMAPs. You do not report intake of energy, carbohydrates, fat and protein. Did you measure them, if not, why not? If you did, please report.
I did not measure the amount of carbohydrates, fat and protein because it was not related to the aims of the study.
One can argue that the diet may have become a low calorie diet unintentionally, by doing so reduced symptoms. Did you weigh your patients at the start and at the end? If weight was reduced on average, this might explain extraordinary results.
I weighed my patients at the beginning and at the end of the diet, but their weight remained stationary or it has had very small insignificant variations enough to explain the results.
Did instruct restraining from probiotics, peppermint oil capsules, fiber supplements such as psyllium, or recommend reducing caffeine, bubbly drinks and fat consumption? Did you monitor the use of these possible dietary dietary treatment modalities?
I suggested my patients not to follow irritable bowel therapies including probiotics, peppermint oil capsules, fiber supplements such as psyllium. I racommended them to drink caffeine, bubbly drinks and to eat fat food with moderation but I did not monitor this strictly.
Your application of re-introduction is interesting. Please contrast it shortly to the methods recommended by Kings college and Monash team J Hum Nutr Diet. 2018 Apr;31(2):239-255.
L 195-197 I wrote “There are no clear indications on the order with which to test the various FODMAPs. The reintroduction could be made on the basis of personal experiences, tastes and preferences” L 206-209 “If the reintroduction of a specific FODMAP did not cause symptoms it was possible to test other food containing the same FODMAP, or to pass to evaluate a new FODMAP, considering the indications on the test modality.”
In my opinion the reintroduction diet is so similar to other studies and that published in J Hum Nutr Diet. 2018 Apr;31(2):239-255.

Reviewer 3 Report
COMMENTS – Comments and Suggestions for Authors (will be shown to authors)
Keywords: The keywords cannot be the same as those in the title, so those must be changed (Line 36)
Introduction:
- References are missing. Important data is explained which is not referenced, for example, “More precisely, three alterations interacting with each other are the main pathogenetic mechanisms underlying this…”. (Line 45) In addition, several studies are mentioned and only one is referred to. (Line 48)
- Not to be explained in the first person(Line 81)
- Possible hypotheses should be highlighted.
Methods:
- All criteria used must always be referenced (Rome IV criteria, line 85)
- Was the study approved by the ethics committee of the center where the Hepatogastroenterology Unit is located? (Added at line 96)
- The inclusion criteria that were taken into account should be indicated.
- In Table 1, foods or food groups should be paired for ease of reading (Line 109)
- The note written below Table 1 can be indicated within the table, or you must indicate what type of note it is. (Lines 110-111)
- It should be indicated because that order of food reintroduction has been indicated. New methods and protocols should be described in detail. (Line 114-123)
- It should be stated what each test is used for the evaluation of symptoms and compliance with the diet. Some are explained and others not. (Lines 136-149)
- The study had to follow the CONSORT 2010 checklist, so it would be appropriate to indicate this in the study.
Results:
- The results do not have to be listed. (For example lines 171-174)
- All tables should be more explanatory and orderly. You should always indicate which parameter you are talking about and explain it in a note under the table. All abbreviations should be indicated above in the text or in a note under the table (For example, IBS-C ; IBS-D…)
Discussion:
- The introduction to the discussion is very broad and the explanations are not referenced. The results should be discussed and interpreted in the perspective of previous studies and working hypotheses. (Lines 200-224)
- All scales used must always be referenced. (for example, Morisky scale: line 220)
- If you are talking about several studies, you should reference all of them, not just one. (Line 233)
- All limitations of the study should be indicated
Conclusion:
- The conclusion is a continuation of the discussion. Therefore, lines 251-259 could be introduced into the discussion following a consistent line.
- The conclusion should be concise and clear, reflecting in a linked way the most relevant results of the study.
Conflicts of Interest:
- Authors must identify and declare any personal circumstances or interests that may be perceived as inappropriately influencing the representation or interpretation of reported research results. If there is no conflict of interest, please state "The authors declare no conflict of interest."
Author Contributions:
- The contributions that each author has made to the study should be indicated.
Others:
- All abbreviations should be defined in parentheses the first time they appear in the abstract, main text, and in figure or table captions and used consistently thereafter.
Reviewer 4 Report
This is an interesting study which prospectively recruited subjects with IBS symptoms into a FODMAP intervention. The strength is the sample size and the duration of follow-up. The weakness is the lack of a control group.
Major Points:
-it may help the reader to have a flow diagram to illustrate the study design
-The lack of a control group should be mentioned as a limitation in the discussion
Minor points:
-line 77 - spelled or spelt?
-line 194: this sentence on compliance is confusing and could be made more clear
-it appears that reference 7 and 22 are the same
Author Response
Response to Reviewer 4 Comments
I thank the reviewer for the suggestions.
Point 1: It may help the reader to have a flow diagram to illustrate the study design
The lack of a control group should be mentioned as a limitation in the discussion
Response 1: I thank the reviewer for the suggestions.
Line 590 I added a flow diagram to illustrate the study design in the manuscript.
Line 325, 330 I added the suggested limitation.
Point 2:
Line 77- spelled or spelt?
Line 194: this sentence on compliance is confusing and could be made more clear
-It appears that reference 7 and 22 are the same
Response 2: I thank the reviewer for the suggestions.
Line 77 : Line 39 spelt
Line 194: 273-274 Eight patients did not follow completely the restriction phase with good adhesion, the other patients did it.
Round 2
Reviewer 2 Report
The text has imroved now.
L 329-331 A gastroenterologist gave dietary instructions and I wrote it in limitations
This information must also be in the methods. Please, add to the methods too.
In dietary modification studies it is a good practise to control overall dietary changes, especially macronutrients. Please, add the lack of monitoring of overall diet as a limitation.
Reviewer 3 Report
COMMENTS – Comments and Suggestions for Authors (will be shown to authors)
Methods:
- In Table 1, foods or food groups should be paired for EASE OF READING (Line 109)
- The study had to follow the CONSORT 2010 checklist, so it would be appropriate included in the study (and submit).
Results:
- The results are not clearly presented. Authors must improve the order regarding table position in the manuscript.
Discussion:
- All scales used must always be referenced. (For example: Morisky scale in line 220)
- Do not use the first person (singular or plural) (First, second… paragraph)
- Explain in one paragraph all the limitations of the study.
Conclusion:
- Do not use the first person (singular or plural)
- What types of studies would be needed in the future to confirm these results?
Others:
- All abbreviations should be defined in parentheses the first time they appear in the abstract, main text, and in figure or table captions and used consistently thereafter. (For example, IBS)
- Figures: Figure 1 does not have good resolution and should be placed in the main text near first quote
Author Response
Response to Reviewer 2 Comments
Ponit 1:
Methods:
- In Table 1, foods or food groups should be paired for EASE OF READING (Line 109)
- The study had to follow the CONSORT 2010 checklist, so it would be appropriate included in the study (and submit).
Response:
I thank the reviewer for the suggestions.
- In table 1 fodds and foods group are paired now for ease of reading at Line 202
- I compiled CONSORT 2010 checklist, attached as a file
Ponit 2:
Results:
- The results are not clearly presented. Authors must improve the order regarding table position in the manuscript.
Response:
I thank the reviewer for the suggestions. I changed the order of result points
Ponit 3:
Discussion:
- All scales used must always be referenced. (For example: Morisky scale in line 220)
- Do not use the first person (singular or plural) (First, second… paragraph)
- Explain in one paragraph all the limitations of the study.
Response:
I thank the reviewer for the suggestions.
- I used references for every used scales
- I changed all person in second person
- I explained in Discussion paragraph limitations of the study
Ponit 4:
Conclusion:
- Do not use the first person (singular or plural)
- What types of studies would be needed in the future to confirm these results?
Response:
I thank the reviewer for the suggestions.
- I changed all person in second person
- I specified that “We need further studies with a long follow-up like our study or probably even longer and with a great number of enrolled patients to confirm our results to assess the possibility to use this dietary regimen for more than 6 weeks without side effects.” at Line398-402
Ponit 5:
Others:
- All abbreviations should be defined in parentheses the first time they appear in the abstract, main text, and in figure or table captions and used consistently thereafter. (For example, IBS)
- Figures: Figure 1 does not have good resolution and should be placed in the main text near first quote
Response:
I thank the reviewer for the suggestions.
- I revised all defined in parentheses abbreviations as you suggested me
- I made better resolution of Figure 1 and I moved it in the main text near first quote
